# Translation and Validation of Digital Competence Indicators in Greek for Health Professionals: A Cross-Sectional Study

**DOI:** 10.3390/healthcare12141370

**Published:** 2024-07-09

**Authors:** Alexandra Karvouniari, Dimitrios Karabetsos, Christos F. Kleisiaris, Savvato Karavasileiadou, Nadiah Baghdadi, Virginia-Athanasia Kyrarini, Evangelia Kasagianni, Afroditi Tsalkitzi, Maria Malliarou, Christos Melas

**Affiliations:** 1Department of Nursing, School of Health Science, Hellenic Mediterranean University, 71410 Heraklion Crete, Greece; melas@hmu.gr; 2Department of Neurosurgery, University Hospital of Heraklion, 71500 Heraklion Crete, Greece; alobar@pagni.gr; 3Department of Nursing, University of Thessaly, Gaiopolis, 41500 Larissa, Greece; kleisiaris@uth.gr (C.F.K.); malliarou@uth.gr (M.M.); 4Department of Community Health Nursing, College of Nursing, Princess Nourah bint Abdulrahman University, P.O. Box 84428, Riyadh 11671, Saudi Arabia; 5Nursing Management and Education Department, College of Nursing, Princess Nourah bint Abdulrahman University, P.O. Box 84428, Riyadh 11671, Saudi Arabia; nabaghdadi@pnu.edu.sa; 6Department of Mathematics, University of Patras, 26504 Rio, Greece; up1089723@ac.upatras.gr; 7424 General Army Training Hospital, 56429 Thessaloniki, Greece; e.a.kasagianni@army.gr; 8401 General Military Hospital of Athens, 11525 Athens, Greece; a.g.tsalkitzi@army.gr

**Keywords:** digital skills, healthcare professionals, DIGCOMP framework, instrument

## Abstract

Background: it is widely accepted that living in the digital transformation era, the need to develop and update new professional skills and tools in health sectors is crucially important. Therefore, this study aimed to explore the reliability and validity of the Digital Competence Indicators tool in assessing the digital skills of Greek health professionals. Methods: in this cross-sectional study, 494 health professionals, including doctors (175) and registered nurses (319) working in four Greek hospitals were recruited and willingly participated using a convenience-sampling method. The original framework of Digital Competence Indicators was translated from English to Greek based on guidelines for cross-cultural adaptation of questionnaires. The validity of the tool was explored using confirmatory factor analysis (CFA) to verify the fit of the model using inductive techniques. The instrument reliability was confirmed using Cronbach’s alpha (α) and McDonald’s Omega coefficients. Results: the reliability was estimated at 0.826 (Cronbach’s-α) and 0.850 (McDonald’s Omega-ω). The indicators of CFA were all calculated within an ideal range of acceptance. Specifically, the CFA comparative fit index produced the following adjustment indices: x^2^/df = 1.152 (*p* = 0.037), CFI = 0.997, Lewis index (TLI) = 0.966, and root mean square error of approximation (RMSEA) = 0.018. Conclusions: The present study demonstrated that the Digital Competence Indicator instrument has high reliability, internal consistency, and construct validity and, therefore, it is suitable for measuring digital skills of health professionals.

## 1. Introduction

The rapid improvement of technology in the health sector results in the development of new skills for professionals and their continuous updating, which will allow the meeting of contemporary demands [1]. While managing the recent pandemic COVID-19, one of the biggest challenges has been to ensure that health workers have sufficient capabilities and skills to meet the digital demands of healthcare. Weaknesses in health professionals were identified and skills shortages were highlighted [2], especially in digital skills that are considered necessary to respond to the challenges of today’s Digital Revolution through medical diagnosis, decision-making and identification of processes and available resources [3]. Digital competence is a mix of skills, capabilities and behaviors, which health professionals need to develop in order to integrate information technology into their professional careers and improvement [4,5].

Assessing the digital skills of professionals working in the field of health is therefore a need and requirement, while the literature supports the view that existing measurement tools prove particularly poor. In mapping the scientific field, we identified validated tools that have been used in studies such as the Digi-HealthCom [6,7], the e-HEALS tool [8], the Canadian Nurse Informatics Competency Assessment Scale [9], the Pre-test for Attitudes Towards Computers in Healthcare Assessment tools, the Multicomponent Assessment of Computer Literacy [9,10], the Awareness, Knowledge, Attitude, and Skills tool [11], the Health Literacy Questionnaire [12], and the Digital Health Literacy Instrument [13,14]. However, these instruments were designed more to evaluate healthcare professionals’ beliefs and digital literacy than to measure their digital skills. It is worth noting that the majority of researchers focus on studying specific areas for digital skills: for example, digital skills required in digital applications [6,15], digital skills and telemedicine awareness [16], digital skills and correlation with digital stress in psychiatric hospitals [17], and digital skills related to clinical practice nurses [18], using the corresponding measurement scales.

In Greece, to the best of our knowledge, there are no competence-related tools available to assess digital skills of healthcare professionals or, in fact, for workers in a hospital environment including all health professionals and not only specific specialties such as, for example, nurses or physiotherapists, etc. Given the workload that characterizes Greek hospitals due to understaffing of health personnel, the extraneous duties that health workers are charged with, due to a lack of other non-health specialties, and the mostly negative attitude of staff to participate in surveys due to lack of time, we wanted to use a measurement tool that is easy to understand, short to complete, suitable for all health professionals, and which is able to assess the levels of necessary digital skills in their entirety.

Taking into account the particularities of the Greek National Healthcare System, as well as the possibility given by the Digital Competence framework to analyze its structure in distinct dimensions depending on the points of interest [4], we chose to use this framework for our research. Thus, it was possible to isolate from the framework the competency domains and their available indicators, which provide a more simplified overview of the framework, which can be used as a measuring scale in this research. Within a framework, we consider providing an effective validity of the Digital Competence Indicators (domain) to fill this gap. The selection of the tool was based on the fact that it captures digital skills in specific activities and practices, is based on the main ideas of the framework, is comprehensible and concise, and, finally, it seems to better reflect the Greek data. Consequently, the present study aims to translate and validate the selected instrument into the Greek language and test it among health professionals in Greece to measure their digital competences.

## 2. Materials and Methods

### 2.1. Study Design

This study took place in two stages in a cross-sectional design. During the 1st phase, an extensive literature review was carried out in order to search for methods that have been used to measure digital skills. The used methods include surveys that require the use of the internet or online applications, surveys through self-assessment of digital skills and performance tests in controlled environments [19]. We chose to use a questionnaire because of the strong points of this method, such as the ability to present a large number of questions on a wide range of skills in a short period of time, simple scoring, quick processing, cost-effectiveness and cost efficiency [20]. The international literature was then searched for the selection of the instrument as well as its translation and cultural adoption. The second phase includes the statistical processing we performed in order to validate the measurement tool.

This study is a part of the research activities of the Nursing Department of the Hellenic Mediterranean University and, therefore, was ethically approved by the Review Board protocol number (IRB 4634/29-03-2021). Before conducting the study, all necessary permissions were obtained from the respective services and, specifically, by the Scientific Committees of all participating hospitals (protocol No: 5584/1973/26-06-2021). Data collection was carried out in the form of a self-assessment questionnaire, which was voluntarily completed by the participants. Access to the questionnaires was the responsibility of the principal investigator, while the use of the data was carried out exclusively for the research. During the conducting of the study, all ethical principles governing scientific research were observed. The procedures and purpose of the research were explained to the participants, their assent and consent was obtained, and anonymity was respected, as well as the right to voluntarily withdraw from the research at any time.

All those involved in the process of collecting and processing research data are bound by the obligation to comply with the provisions of the General Data Protection Regulation of the European Union (679/2016) for the protection of personal data [21]. Furthermore, the research was conducted in accordance with the Declaration of Helsinki and the elements described by the Code of Ethics for Nurses developed by the ICN (International Council of Nurses, Geneva, Switzerland, 2021) were applied [22].

### 2.2. Participants and Settings

The sample of the study is representative of doctors and registered nurses from four Greek tertiary-care hospitals. Specifically, 251 Hellenic Air Force General Hospital and Athens Naval Hospital in terms of the military, and Asklepieio Voulas General Hospital and Henry Dunant Hospital Center in terms of the civilian hospitals. The minimum sample size for Confirmatory Factor Analysis was calculated as equal to 289, according to R package lavaan [23]. Participation in the research presupposed the fulfillment of certain criteria, namely, (a) the participants had to work in the hospital with the professional status of doctor or nurse, (b) the nurses had to be registered nurses (European Qualification Framework—EQF 6), and (c) the participants had to be willing to take part in the research. Exclusion criteria from the study were the following: (a) students in hospitals, (b) practitioner nurses (EQF 5), and (c) health professionals with specialties other than doctors and nurses. No subjects were excluded from the study, as all met the criteria we set.

We recruited the sample from different types of hospital units during their staff meetings. Access to the participants was a responsibility of the principal investigator, while the use of the data was carried out exclusively for the purposes of the research. Consent was obtained from all participants, after receiving explanations of the objectives of this study as well as clarifications for completing the questionnaire from researchers. In addition, all participants were informed that they could withdraw from the study at any time. The time period for data collection lasted from June through September 2021, using the convenience sampling method.

### 2.3. Instrument

As a tool for the study, we used a questionnaire which, in the first part, includes questions about the demographics of the participants and, in the second part, questions on the self-assessment of health professionals’ computer skills.

The European Union, wanting to create a common umbrella of digital competencies for its citizens, created the European Digital Competence Framework, commonly referred to as DigComp [24,25]. DigComp includes five dimensions and its structure can be quite complex, but, because of this complexity, we were able to isolate the domains of interest, specifically the competence areas with their respective indicators. Thus, the original text with the available competency indicators was translated from English to Greek.

In this light, 26 items were chosen, so that each skill is signified by one item. The skills are divided into five competency categories that refer to information, communication, content creation, problem-solving and security. Health professionals are asked to choose the item that reflects the degree to which they use computers for each activity, following a Likert scale consisting of five points, ranging from 1, representing not at all, to 5, representing very much.

In addition, for the tool’s external validation, the variables that can be studied for their correlation with the levels of digital skills were sought through the literature review. The results of the review indicate that factors associated with levels of digital skills are age [26,27,28,29], type of profession [26], gender [28,29], professional experience [26,27], and educational level [28,29]. However, it is worth noting that, although studies show that demographic factors indicate a large number of non-significant determinants, they do emerge as important factors for technical and informational digital skills with high frequency [27]. In this context, the questionnaire included 11 items related to demographic information of the research population. The sociodemographic variables were the following: gender, age, education level, profession, specialty, faculty, department of the working hospital, duration of the specific profession in years, period of work in the specific hospital in years, duties in an administrative position and possession of a computer literacy certificate.

### 2.4. Translation Procedure

A translation procedure and a cultural adoption of the digital skills scale were performed, according to the cross-cultural adaptation of self-report measure guidelines [30]. The term “cross-cultural adaptation” includes a process that can be used when a questionnaire is prepared for use in a different setting, in order to address issues of linguistic as well as cultural adaptation. The method described by Beaton et al. recognizes a total of six stages: (a) initial translation, (b) synthesis of the translations, (c) back translation, (d) pre-final form, (e) pre-testing and (f) final form [31].

At stage 1, two translations from the original language (English) to the target language (Greek) were carried out by two bilingual translators, in particular, by a medical doctor with experience in translating medical books written in English and by a ministry administrative officer with professional experience in managing English documents.

In the next stage, results of the previous translations were composed by the two translators and a common translation was produced. Subsequently, the common translation was back-translated by a third translator, this time from Greek to English. This person was a Greek medical doctor, translator and author of Greek and English books (stage 3). The specific stage is a validation process to confirm that the content of the translated version reflects content identical to that of the original version.

During the fourth stage of the process, the three translators, in collaboration with three other researchers (a medical doctor and two registered nurses with excellent knowledge of English) compared the back-translation form with the original one. After completing a review of the translations that resulted from the above stages, these six translators, after collaborating, came up with the unification of the scale, forming its pre-final form.

The fifth stage was pre-testing the scale before it took its final form. The trial was carried out on 30 healthcare professionals (medical doctors and registered nurses working in hospitals), to whom the scale was distributed in order to explore the understanding of its data and to identify possible incomprehensible expressions that may have influenced the collection of the study’s primary data.

Following the above steps, the final form of the scale was agreed upon by the translators as the last phase in the translation procedure. In essence, this is the result of all the iterations described above. The few differences found among all the forms of the scale that resulted from the mentioned stages led to the last arrangements of the Greek translation, for which all the translators involved collaborated. Through the final stage, the whole process was checked to verify that the recommended steps mentioned in the guidelines for the intercultural process used to adapt a questionnaire had been respected.

It should be noted that although the procedure performed provides some qualitative index of content validity as well as some useful information about how the individual interprets the scale items, it does not offer a valid and reliable measure of the scale, which is performed with additional tests throughout further statistical analysis.

### 2.5. Data Analysis

Data were analyzed with IBM SPSS (Statistical Package for Social Sciences) version 25 software and Jamovi software (version 2.4.11), which is a free, open-source, standalone software offering a point-and-click interface for R scripts. We used descriptive statistics to characterize subjects, as well as the scale distribution of DigComp scores. Numbers and percentages were used to express categorical variables, and mean standard deviation was used to express continuous variables.

Evaluation of the construct validity of the measurement tool was carried out using factor analysis. Confirmatory factor analysis (CFA) was used to verify the fit of the model to the population under study. As our hypothesis is based on a framework, CFA determines which variables will be related to which factors, as well as which factors will be related to each other. In all analyses, model fit was evaluated with standard indices of model fit, namely the comparative fit index (CFI), standardized root means square residual (SRMR), and root mean square error of approximation (RMSEA). A CFI index greater than 0.90 is acceptable, and greater than 0.95 is desirable. The RMSEA index takes values between 0 and 1, where a value of 0 indicates that the model exhibits perfect fit. The RMSEA should be less than 0.08. The significance level for accepting or rejecting the hypotheses was set at a p coefficient of 0.05. Finally, the index SRMR should be <0.08.

Instrument reliability, which refers to the measurement results of a scale rather than the scale itself, was based on internal consistency in order to evaluate the homogeneity of the scale’s questions [32]. Cronbach’s alpha (α) and McDonald’s Omega (ω) coefficients were used to define the internal consistency validity of the digital skills scale. We chose to use Cronbach’s alpha index, as it is a measure of reliability, widely used in many research studies to objectively measure the reliability of a measurement tool. In the case of developing a new measurement scale, reliability coefficient values exceeding 0.7 are usually considered acceptable and indicate that the items exhibit interdependence and homogeneity with respect to the scale being measured. Values above 0.80 are desirable and acceptable for clinical applications [33].

## 3. Results

### 3.1. Participants’ Characteristics

The sample was composed of medical doctors and registered nurses from two military and two civilian hospitals. Out of the total of 494 participants, 291 were female (58.9%) and 203 (41.1%) were male. Their mean age was 38.7 (10.04). Table 1 presents the demographic profile of the study participants. A total of 175 (35.43%) participants were medical doctors and 319 (64.57%) were registered nurses. Overall, 297 (60.12%) had only a BSc degree, while 166 (33.6%) had, in addition, an MSc degree and 31 (6.28%) a PhD degree. In addition, Table 2 shows the distribution of responses for each scale item expressed in percentages. Finally, Table 3 lists the results of the correlations between the subscales of the measurement tool. No statistically significant correlation was found between the subscales of the questionnaire.

### 3.2. Construct Validity

From the validation of the factorial structure performed by CFA, a five-factor model was conducted. Table 4 and Table 5 list the results obtained from the construct validity, all of which are within acceptable limits. The fit indices resulting from the CFA are as follows: χ^2^/df = 1.152 (*p* = 0.038), comparative fit index (CFI) = 0.997, Tucker–Lewis index (TLI) = 0.966, and root mean square error of approximation (RMSEA) = 0.018. These excellent results for all calculations indicate that the construction of the 5-factor measurement scale which we proposed for healthcare professionals could be accepted.

### 3.3. Reliability

With respect to scale reliability, the following Cronbach’s alpha values were observed: the Cronbach’s alpha value for all items of the scale was 0.823, as well as greater than 0.7 for each item of the scale separately (Table 6). More specifically, the Cronbach’s alpha value was 0.950 for factor 1 (Information), 0.949 for factor 2 (Communication), 0.956 for factor 3 (Content creation), 0.966 for factor 4 (Problem-solving), and 0.920 for factor 5 (Safety), as shown in more detail in Table 7.

## 4. Discussion

In this study, we demonstrate that the concepts incorporated in the measurement tool can be applied to the health staff of Greek hospitals, representing a validated and objective tool to measure digital skills for health professionals, which is considered beneficial and necessary. It is worth mentioning that we found no studies conducted on health professionals using DigComp worldwide to compare our results.

In contrast to our study, which validates a tool that measures the digital skills of health professionals, other studies validate tools that measure digitalization skills and digital health [34], electronic health literacy levels [8], information technology, and computer skills [16] of health professionals. Also, while our measurement tool focused on digital skills per se, other tools measure the digital skills required for certain specialized digital technologies such as telemedicine [12,35] and digital applications [15]. Regarding the specialty of health professionals to which the measurement tool is addressed, our tool is addressed to both doctors and nurses. The literature mentions several tools specific to nurses such as the Canadian Nurse Informatics Competency Assessment Scale [9], tools for the digital skills of clinical practice nurses [18], the Nursing Informatics Competency-Based Assessment for Nursing Personnel in Primary Healthcare [36], and the Nursing Informatics Competency Assessment Tool [37].

However, studies that used a very similar instrument to the one in the present study, as regards the digital skills of health professionals, have shown comparative results. In particular, e-health competence does not only depend on personal digital skills, but also on the development of new work processes and practices [38]; teaching digital literacy skills can be performed through didactic and experiential approaches [39], while internet use and e-health literacy of health care professionals seems to be significantly good [26]. Information literacy and digital competence are considered essential for managing the volume of information available [40].

As a result of our research, we could find only a few surveys carried out in Greece among health professionals which investigate the digital skills between doctors and nurses and the differences in their levels. To the best of our knowledge, the general findings are that doctors, compared to nurses, have higher levels of knowledge in topics that require digital skills, such as big data [41], as well as knowledge about the use of Information and Communication Technologies in the field of their work [42].

Further studies focus on the critical role that education and learning digital skills play. Special emphasis is placed on training health professionals, due to their modest levels of technological skills [43], training undergraduate healthcare students for both the safe and effective use of electronic patient records [44], and for the development of digital literacies in general [45]. The majority of researchers stress the need to integrate the learning of digital skills into the curriculum of healthcare professionals [46], as well as continuing education in the healthcare workplace [47]. Likewise, some indicate that teaching computer programming to hospital doctors and nurses is possible, providing them with an introductory skill level [48], while the development of an online learning resource appears to be useful [48].

Moreover, the data studied reveal that digital skills are directly related to both policy-making and educational planning of health professionals. This fact is vital for a society that is within the Knowledge Society. Thus, taking into consideration the existing relevant studies, testing new tools that prove to be valid and reliable, as well as relevant for the study of digital skills in the field of health, can be useful. In addition, these tools need to be adapted to the context in which they are used.

### 4.1. Future Implications

We believe that the questionnaire of the current research is a reliable and valid measurement tool which will be useful and can be easily used in future research studies. It can also be tested in other health professional specialties besides doctors and nurses, such as pharmacists, physiotherapists, etc., as well as proving useful in other countries with similar hospital environments to those of Greece. The results of these studies may prove particularly beneficial for health policy makers who make decisions about measures and interventions needed to provide quality healthcare services, such as improving the digital skills of health personnel.

Additionally, the tool used is a starting point for the development of similar tools that can be used by health professionals. Perhaps the greatest challenge facing any healthcare system is finding ways to adapt to the growing pace of change. What can be done for patients is also changing, with the development of new and advanced technological interventions. Some of them require new skills. Health professionals play a crucial role in meeting these challenges through the many decisions they make about diagnosing and treating patients and determining both the procedures and the resources that will be used.

### 4.2. Limitations

The present study is affected by some limitations. First, the participants were intentionally sampled from hospitals in the prefecture of Attica and not from hospitals located in the wider region of Greece, and, therefore, the above restriction may affect the generalization of the results. Second, in the measurement tool test, the participating nurses were all registered nurses (European Qualification Framework—EQF 6) and not practitioner nurses (EQF 5), and thus this tool is validated and reliable only for professional nursing staff. Third, the participants self-assessed, and as the questionnaire is a measurement method that carries the risk of subjectivity, this may lead to an overestimation or underestimation of the skills possessed by the study population.

## 5. Conclusions

In conclusion, the current research revealed that the study questionnaire based on the DigComp framework seems to be a proven reliable and valid instrument, usable to investigate the digital skills of healthcare professionals. This self-assessment, easy-to-use instrument can be completed quickly and seems to provide a brief and effective assessment of the digital skills of hospital doctors and nurses. It is planned to be used by the researchers of this article in a larger population sample to investigate the digital skill levels of health professionals in other military and civilian hospitals, as well as the factors associated with them. Similarly, it can be used in the future by other researchers as a tool that fulfills quite strong characteristics. In the hospital environment, dominated by an increased workload and minimal time available for consultations, the short completion time of an instrument is crucial. Overall, the results of the present study demonstrate that the concepts incorporated in the questionnaire can be feasibly applied to Greek health professionals.

## Figures and Tables

**Table 1 healthcare-12-01370-t001:** Demographic profile of participants (n = 494).

		n	n%
Gender	Female	291	58.91%
	Male	203	41.09%
Faculty	Medical doctor	175	35.43%
	Registered nurse	319	64.57%
Educational level	BSc	297	60.12%
	MSc	166	33.60%
	PhD	31	6.28%
Age ^1^	38.7 (10.04)		

^1^ Age expressed as mean (standard deviation).

**Table 2 healthcare-12-01370-t002:** Percentage distribution of responses by instrument scale item.

	Not at All	A Little Bit	Partly	Enough	Very Much
	n%	n%	n%	n%	n%
item 1	19.64%	22.47%	19.03%	20.85%	18.02%
item 2	19.84%	21.86%	21.05%	17.21%	20.04%
item 3	20.04%	20.85%	21.46%	20.04%	17.61%
item 4	21.46%	19.23%	22.87%	18.42%	18.02%
item 5	19.03%	22.47%	20.65%	18.42%	19.43%
item 6	21.46%	20.85%	22.87%	18.02%	16.80%
item 7	21.46%	22.47%	21.05%	18.42%	16.60%
item 8	19.84%	21.66%	20.04%	22.06%	16.40%
item 9	23.48%	20.24%	21.66%	19.23%	15.38%
item 10	22.47%	20.85%	19.64%	22.27%	14.78%
item 11	20.04%	20.04%	21.46%	17.00%	21.46%
item 12	22.67%	17.21%	18.83%	21.05%	20.24%
item 13	20.45%	19.43%	18.22%	20.45%	21.46%
item 14	22.06%	20.45%	18.42%	17.41%	21.66%
item 15	20.65%	20.45%	20.65%	17.00%	21.26%
item 16	19.84%	19.23%	21.05%	23.08%	16.80%
item 17	18.02%	21.26%	22.27%	21.26%	17.21%
item 18	17.41%	19.43%	24.90%	21.46%	16.80%
item 19	20.45%	20.04%	19.43%	21.26%	18.83%
item 20	18.42%	20.85%	22.27%	19.23%	19.23%
item 21	19.23%	20.24%	23.89%	18.02%	18.62%
item 22	20.04%	20.24%	22.67%	18.42%	18.62%
item 23	17.61%	21.05%	20.45%	22.27%	18.62%
item 24	20.04%	20.04%	21.46%	20.45%	18.02%
item 25	19.03%	20.04%	19.23%	20.65%	21.05%
item 26	17.00%	22.47%	18.83%	21.66%	20.04%

**Table 3 healthcare-12-01370-t003:** Correlations between the subscales of the questionnaire.

		Information	Communication	Content Creation	Problem-Solving	Safety
Information	Pearson Correlation	1	0.046	−0.006	0.018	0.007
	Sig. (2-tailed)		0.175	0.862	0.590	0.827
	N	857	857	857	857	857
Communication	Pearson Correlation	0.046	1	−0.025	−0.061	−0.005
	Sig. (2-tailed)	0.175		0.473	0.077	0.888
	N	857	857	857	857	857
Content creation	Pearson Correlation	−0.006	−0.025	1	0.011	−0.009
	Sig. (2-tailed)	0.862	0.473		0.751	0.795
	N	857	857	857	857	857
Problem- solving	Pearson Correlation	0.018	−0.061	0.011	1	0.059
	Sig. (2-tailed)	0.590	0.077	0.751		0.082
	N	857	857	857	857	857
Safety	Pearson Correlation	0.007	−0.005	−0.009	0.059	1
	Sig. (2-tailed)	0.827	0.888	0.795	0.082	
	N	857	857	857	857	857

**Table 4 healthcare-12-01370-t004:** Test for Exact Fit.

χ^2^	df	*p*
333	289	0.038

**Table 5 healthcare-12-01370-t005:** Fit Measures.

RMSEA 90% CI				
CFI	TLI	RMSEA	Lower	Upper
0.997	0.966	0.0176	0.00457	0.0256

**Table 6 healthcare-12-01370-t006:** **Full-scale** reliability results.

	Corrected Total-Item Correlation	Cronbach’s Alpha if Item Deleted
item 1	0.322	0.819
item 2	0.292	0.820
item 3	0.321	0.819
item 4	0.274	0.821
item 5	0.352	0.818
item 6	0.257	0.821
item 7	0.235	0.822
item 8	0.240	0.822
item 9	0.233	0.822
item 10	0.223	0.823
item 11	0.292	0.820
item 12	0.240	0.822
item 13	0.276	0.821
item 14	0.282	0.821
item 15	0.252	0.822
item 16	0.615	0.807
item 17	0.549	0.810
item 18	0.498	0.812
item 19	0.494	0.812
item 20	0.557	0.809
item 21	0.562	0.809
item 22	0.558	0.809
item 23	0.518	0.811
item 24	0.524	0.811
item 25	0.130	0.826
item 26	0.152	0.825

**Table 7 healthcare-12-01370-t007:** Reliability results for the five factors of the scale.

	Corrected Total-Item Correlation	Cronbach’s Alpha if Item Deleted
Factor 1: Information	α = 0.950	
item 1	0.952	0.922
item 2	0.852	0.940
item 3	0.838	0.942
item 4	0.829	0.944
item 5	0.840	0.942
Factor 2: Communication	α = 0.949	
item 6	0.957	0.919
item 7	0.835	0.941
item 8	0.837	0.940
item 9	0.843	0.940
item 10	0.825	0.943
Factor 3: Content creation	α = 0.956	
item 11	0.958	.931
item 12	0.849	.950
item 13	0.857	.948
item 14	0.865	.947
item 15	0.852	.949
Factor 4: Problem-solving	α = 0.966	
item 16	0.979	0.957
item 17	0.840	0.963
item 18	0.834	0.963
item 19	0.841	0.963
item 20	0.847	0.963
item 21	0.839	0.963
item 22	0.835	0.963
item 23	0.848	0.963
item 24	0.842	0.963
Factor 5: Safety	α = 0.920	
item 25	0.851	.
item 26	0.851	.

## Data Availability

The data that support the findings of the present study are available on request from the corresponding author.

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
