# Peer review of "Translation and Validation of Digital Competence Indicators in Greek for Health Professionals: A Cross-Sectional Study"

_healthcare, 2024, doi:10.3390/healthcare12141370_

Round 1

Reviewer 1 Report

Comments and Suggestions for Authors

The article is interesting and valuable. The article is important to collegous working in the field of medicine and bioengineering. The introduction state the purpose of the paper. The number of references is proper and references are modern. The authors hypothesize that the DigComp questionnaire is a useful tool for assessing the digital skills of healthcare workers.They prove it in a logical and scientific manner. The research group is impressive. The discussion is interesting. 

Major:

Authors should clearly articulate the found research gap from a scientific perspective in the introduction section.

No new scientific method or approach invented by the authors for statistical evaluation.

No formulas used by the statistical software in the text of the article.

The authors should indicate the direction of further research in the conclusions.

Minor:

The paragraps 2.1 , 2.2, 4.1, 4.2 are too short. You should reedit the text.

You should add at least 5 modern references to introduction section. 

Conclusions: 

The work is weak from a scientific point of view. The idea is very successful. The article has great practical value. Congratulations. Work is going in the right direction. Indication for publication in Healthcare is positive.

Author Response

Dear Reviewer 1,

Major:

Comments 1: Authors should clearly articulate the found research gap from a scientific perspective in the introduction section.

Response 1:

  • Thank you so much for your valuable suggestions.
  • We agree with your points that further elaborating on this point using new data would be helpful.
  • We rewrote the introduction section (page 2) following your suggestions.
  • In detail:
  1. In the 2nd paragraph (original manuscript, page 2, lines: 51-59) we added more data from the international literature with 8 new recent literature references (references: 7, 12, 13,14,15,16,17 and 18) about existing measurement tools that have been used in similar studies.
  2. The new text added is in the revised manuscript in page 2, 2nd paragraph, lines 57-63 and the new amended 2nd paragraph is listed in the revised manuscript on page 2, lines 50-63.
  3. In the revised manuscript the above new literature references are mentioned in the list of references, as cite [7] on page 11, lines: 401-403 and cites [12-18], on page 12, lines: 413-431.
  4. We have rewritten the former 3rd paragraph (original manuscript, page 2, lines: 60-72), which is now formed into two new paragraphs, and are in the revised manuscript the new 3rd paragraph (page 2, lines: 64-73) and the new 4th paragraph (page 2, lines: 74-86) of introduction section.
  5. The data we have added gives more details about the gap that exists in Greece in measuring the digital skills of healthcare professionals and the reasons that led us to choose the specific measurement scale to support the above gap.

Comments 2: No new scientific method or approach invented by the authors for statistical evaluation.

Response 2:

  • Thank you so much for your valuable comments.
  • Yes indeed, we did not invent a new scientific method for statistical evaluation. We chose to use from an already existing conceptual framework (the DIGCOMP) the available indicators of digital competence, using them as a measurement scale, following the process of translation and statistical analysis, with the evaluation of the validity and reliability of the measurement tool.

 Comments 3: No formulas used by the statistical software in the text of the article.

Response 3:

  • Thank you so much for your valuable comments.
  • According to our statistician, the entire statistical analysis methodology is presented in the subsection titled data analysis. In case something more is needed, please provide us an example.

Comments 4: The authors should indicate the direction of further research in the conclusions.

Response 4:

  • Thank you so much for your valuable suggestions.
  • We agree with your points that further elaborating on this point using new data would be helpful.
  • We have added to the conclusions section (original manuscript, page 10, one paragraph, lines: 328-336) additional information about the direction of further research. The new text added is in the revised manuscript in the conclusions section on page 10, lines: 345-349. The revised conclusion section on page 10 consists of one paragraph, lines 356-368.

Minor:

Comments 5: The paragraphs 2.1, 2.2, 4.1, 4.2 are too short. You should reedit the text.

Response 5:

  • Thank you so much for your valuable suggestions.
  • We agree with your points that further elaborating on this point using new data would be helpful.
  • We rewrote all the above subsections following your suggestions.
  • Specifically, regarding subsection 2.1:
  1. We rewrote the subsection 2.1 “Study Design” (original manuscript, page 2, one paragraph, lines: 75-77).
  2. In the revised manuscript subsection 2.1 “Study Design” is now on pages 2 and 3 and consists of three paragraphs:

1st paragraph (page 2-3, lines: 89-100),

2nd paragraph (page 3, lines: 101-112) and

3rd paragraph (page 3, lines: 113-118).

  1. We added two new literature references to the text above (1st paragraph, page 2-3, lines: 89-100), cited as [19] and [20] (page 12, lines: 432-435).
  • Regarding subsection 2.2:
  1. We have added additional information about participants and settings in the subsection 2.2 (original manuscript, page 2, one paragraph, lines: 79-86).
  2. In the revised manuscript subsection 2.2 is on page 3 and formed into 2 paragraphs. The 1st new paragraph is set out in lines 103-110 and the new 2nd paragraph is set out in lines 111-118.
  • Regarding subsection 4.1:
  1. We have added additional information in the subsection 4.1 “Future Implications” (original manuscript, one paragraph, page 10, line 311) and specifically we added as a 1st paragraph more information about the significance of our study findings and possible future implications.
  2. So, we added a new 1st paragraph in the revised manuscript (page 10, lines: 329-336) and subsection 4.1 “Future Implications” is now on page 10 and formed into 2 paragraphs (page 10, lines 329-344).
  • Regarding subsection 4.2:
  1. We have added additional information in the subsection 4.2 “Limitations” (original manuscript, one paragraph, page 10, lines: 320-326) and specifically we added at the end of the paragraph, a sentence with an additional limitation that we had in our research.
  2. In the revised manuscript the sentence we added in subsection 4.2 “Limitations” is on page 10 on lines 352-354, so the modified paragraph of subsection 4.2, is now on page 10, in lines 346-354.

Comments 6: You should add at least 5 modern references to introduction section. 

Response 6:

  • Thank you so much for your valuable suggestions.
  • We agree with your points that further elaborating on this point using new data would be helpful.
  • We have added 8 new recent literature references cited as [7], [13],[14],[15],[16],[17] and [18] in the introduction section, giving more data from the international literature on existing measurement tools that have been used in similar studies.
  • In the revised manuscript the above new literature references are mentioned in the list of references, namely:

Reference 7, page 11, lines: 401-403.

Reference 12, page 12, lines: 413-415.

Reference 13, page 12, lines: 416-417.

Reference 14, page 12, lines: 418-419.

Reference 15, page 12, lines: 420-422.

Reference 16, page 12, lines: 423-426.

Reference 17, page 12, lines: 427-429.

Reference 18, page 12, lines: 430-431.

Comments 7: Conclusions: The work is weak from a scientific point of view. The idea is very successful. The article has great practical value. Congratulations. Work is going in the right direction. Indication for publication in Healthcare is positive.

Response 7:

  • Thank you so much for your positive comments.
  • Yes indeed, we realize that we did not develop a new measurement tool of our own which clearly has more scientific value in a study. Taking into account the ever-increasing demands created in the health field for the use of new technologies and services that presuppose digital knowledge, especially after the COVID pandemic, we chose for a shortcut, to use ready-made data that has been tested on the field of digital skills. Our goal was to find a comprehensive questionnaire that can be applied to health professionals.

We appreciate the thoughtful and positive comments from the Academic Editor and the Reviewers. We hope you find the revised manuscript suitable for publication. Thank you once again for your effort and time.

Reviewer 2 Report

Comments and Suggestions for Authors

Considering the current high development of digital health care demands, the present article is of utmost interest and worth  for publishing.
Aware of the increasing utility of digital competencies, the European Union created a European Digital Competence Framework-DigComp,  as a model to be spread and promoted in European  countries.
This article aimed in the first phase at selecting the right instrument  based on  DigComp to be used and translating it into the Greek language, and in the se-cond phase at validating this tool with doctors and registered nurses from four Greek tertiary hospitals. The complex method of work is presented clearly with all necessary  comprehensive details. The large questionnaire  ,translated in Greek language  ,with 26 items, divided into 5 chapters (information, commu-nication, content creation, problem-solving, and security ), was completed by 494 persons serving for exploratory and confirmatory factor analysis, being finalized with reliability measurements, all based on specific statistical evalua-tions with proper index calculations (Kaiser-Meyer-Olkin, Bartlett,-Cronbach alpha, Mc Donald’s Omega, also). The rigorous analysis of the obtained data led to the conclusion that digital skills are directly connected to policymaking and educational planning of health professionals; these skills are mandatory in a developing world with increasing  complexity of exploration , medical diagnosis , treatment and complex monitoring procedures.

Comments
The study is useful, as in real life  in Greece the proportion of participants in the study in about 40% of cases miss almost totally the usage of computers, with only about 20 % being skilled in this field. Or, it is of decisive importance to know  the quality of the experience of the medical personnel before starting the training for technological skills. Therefore, I would recommend to  present concretely the specific situation of strengths and weaknesses in the group of doctors versus nurses  in order to adapt the training to the real demands of each group in its daily activity; that could assure a new valid, reliable, relevant and adapted digital skill for each group of subjects.

Correction
-the references should respect the recommendations regarding….year, volume, range of pages (like references 1,2,5,6,9,16,17, 33, ,36,37)

Author Response

Dear Reviewer 2,

Reviewer: Considering the current high development of digital health care demands, the present article is of utmost interest and worth for publishing.

Aware of the increasing utility of digital competencies, the European Union created a European Digital Competence Framework-DigComp, as a model to be spread and promoted in European countries.

This article aimed in the first phase at selecting the right instrument based on DigComp to be used and translating it into the Greek language, and in the second phase at validating this tool with doctors and registered nurses from four Greek tertiary hospitals. The complex method of work is presented clearly with all necessary comprehensive details. The large questionnaire, translated in Greek language, with 26 items, divided into 5 chapters (information, communication, content creation, problem-solving, and security), was completed by 494 persons serving for exploratory and confirmatory factor analysis, being finalized with reliability measurements, all based on specific statistical evaluations with proper index calculations (Kaiser-Meyer-Olkin, Bartlett,-Cronbach alpha, Mc Donald’s Omega, also). The rigorous analysis of the obtained data led to the conclusion that digital skills are directly connected to policymaking and educational planning of health professionals; these skills are mandatory in a developing world with increasing  complexity of exploration , medical diagnosis , treatment and complex monitoring procedures.

Response 1:

  • Thank you so much for taking the time to assess our manuscript and for your valuable comments.
  • We also thank you for finding the subject of our manuscript interesting and presenting a summary of our study through your comments, highlighting key concepts we used for our study design, aim, and methodology.
  • We will make all the necessary corrections required in our manuscript as per your recommendations, carefully following your suggestions and the points you highlight to us, in your comments below.

Comments 2:

The study is useful, as in real life in Greece the proportion of participants in the study in about 40% of cases miss almost totally the usage of computers, with only about 20 % being skilled in this field. Or, it is of decisive importance to know the quality of the experience of the medical personnel before starting the training for technological skills. Therefore, I would recommend to present concretely the specific situation of strengths and weaknesses in the group of doctors versus nurses in order to adapt the training to the real demands of each group in its daily activity; that could assure a new valid, reliable, relevant and adapted digital skill for each group of subjects.

Response 2:

  • Thank you so much for your valuable suggestions.
  • We agree with your points that further elaborating on this point using new data would be helpful.
  • We have added a new paragraph with data that we managed to find after searching the existing literature, on the situation observed in Greece regarding digital skills among doctors and nurses.
  • The new paragraph added is in the revised manuscript in section 4. “Discussion”, on page 9, new 4th paragraph, lines: 306-311, with 2 more literature references.
  • In the revised manuscript the above new literature references are cited as: [41] and [42] and mentioned in the list of references, in page 13, lines: 479-482.

Correction 3:

-the references should respect the recommendations regarding….year, volume, range of pages (like references 1,2,5,6,9,16,17, 33, ,36,37)

Response 3:

  • Thank you so much for your valuable suggestions.
  • Following your recommendations, we have re-checked the bibliographic references we have used in our manuscript to be in accordance with the "guidelines to authors" given by the journal.

We appreciate the thoughtful and positive comments from the Reviewers. We hope you find the revised manuscript suitable for publication.

We would like to thank you again for your effort, as well as for taking the time to review our manuscript.

Reviewer 3 Report

Comments and Suggestions for Authors

Reviewer Comment

The digitalizing world is important for the health sector, like all sectors. This importance has gained more importance in the COVID 19 epidemic. Therefore, this study is thought to be important. Although the subject is important, some arrangements are required in the writing system of the manuscript. The summary should be arranged formally. Definitive statements such as "There is no such study" in the introduction section should be avoided. Inclusion and exclusion criteria information about the participants should be added to the method section. It should be explained how the sample size was determined. The discussion section should be detailed and expanded by relating it to different similar studies. The proposed regulations are detailed below. Please make all editing suggestions carefully.

Revision

1.     Page 1, line 19; In the Abstract section, delete the numbers you wrote for each section. Such a notation is not used in the Abstract.

2.     Although you provide reliability results in the conclusion section of the manuscript, this is not mentioned in the Abstract, title and method. This information can be added to these sections.

3.     Page 2, line 60-61; “In Greece, there are no competence-related tools available to assess digital skills of healthcare professionals” It is not appropriate to use expressions such as "there is no certainty" in article writing. It is always possible that you may not be able to access these studies. For this reason, use blunt expressions such as "To the best of our knowledge or as a result of our research, we could not find such a study."

4.     The study's inclusion and exclusion criteria were not specified. These should be explained in the pasticipant section. It should be stated which statistical reference was used to determine how many people will be included in the study. It should be stated whether a power analysis was performed or whether 5-10 times the number of questions were used as a reference. It should be noted if there were people excluded from the study. Additionally, the flow diagram should be added as a figure.

5.     Please indicate in the survey description section what the increase or decrease in the score of the survey used means and whether there is a cut-off score.

6.     Page 5, line 201;2.6. Ethical Considerations” You do not need to use this heading in the method section. Explain this information in the Study design section. When making an ethical declaration, include in this section the information that the research was conducted in accordance with the Declaration of Helsinki.

7.     Page 9, line 278-282; According to the available data we derived from the literature review, the present study is the first in Greece providing a validated digital skills competence instrument for health professionals. Research on the digital skills of employees in various professional fields is constantly gaining ground, but in the hospital environment, it appears to be particularly limited.” There is no need for statements stating the purpose of the study again in the first paragraph of the Discussion section. The beginning of the discussion section should be a paragraph expressing the general result of the study.

8.     Page 9, line 277; The discussion section is short and should be improved. Although it has been stated that there are no similar studies, associations can be made with similar studies. Therefore, the discussion should be reviewed from this perspective. Indirect associations should be made.

Comments on the Quality of English Language

Although the English spelling is understandable, minor editing is required.

Author Response

Dear Reviewer 3,

Reviewer Comment: The digitalizing world is important for the health sector, like all sectors. This importance has gained more importance in the COVID 19 epidemic. Therefore, this study is thought to be important. Although the subject is important, some arrangements are required in the writing system of the manuscript. The summary should be arranged formally. Definitive statements such as "There is no such study" in the introduction section should be avoided. Inclusion and exclusion criteria information about the participants should be added to the method section. It should be explained how the sample size was determined. The discussion section should be detailed and expanded by relating it to different similar studies. The proposed regulations are detailed below. Please make all editing suggestions carefully.

Response:

  • Thank you so much for your valuable comments. We agree with all your suggestions.
  • We will make all the necessary arrangements that are required in the writing system of our manuscript, according to your recommendations, carefully following the instructions you provide us, the points that you highlight to us, as well as the regulations you mention in your comments below.

Revision:

Comments 1: Page 1, line 19; In the Abstract section, delete the numbers you wrote for each section. Such a notation is not used in the Abstract.

Response 1:

  • Thank you so much for your valuable suggestions.
  • Following your instructions, we deleted all the numbers we had written for each section in the Abstract (original manuscript, page 1) and specifically we deleted number 1 on line 19, number 2 on line 22, number 3 on line 29, and number 4 on line 34.

Comments 2: Although you provide reliability results in the conclusion section of the manuscript, this is not mentioned in the Abstract, title and method. This information can be added to these sections.

Response 2:

  • Thank you so much for your valuable suggestions.
  • Following your recommendations, we have added more information on the reliability of the measurement tool in the method section.
  • Please see the added text: “We chose to use Cronbach's alpha index, as it is a measure of reliability, widely used in many research studies to objectively measure the reliability of a measurement tool” in subsection 2.5. “Data Analysis”, page 5, 3rd paragraph, lines: 229-231.

Comments 3:      Page 2, line 60-61; “In Greece, there are no competence-related tools available to assess digital skills of healthcare professionals” It is not appropriate to use expressions such as "there is no certainty" in article writing. It is always possible that you may not be able to access these studies. For this reason, use blunt expressions such as "To the best of our knowledge or as a result of our research, we could not find such a study."

Response 3:

  • Thank you so much for your valuable suggestions. We agree with this comment.
  • Following your recommendations, in the revised manuscript in the “Introduction” section we have added the phrase "To the best of our knowledge" to the sentence you indicated, which now reads:

“In Greece, to the best of our knowledge, there are no competence-related tools available to assess digital skills of healthcare professionals” (page 2, 3rd paragraph, line 64).

Comments: 4.     The study's inclusion and exclusion criteria were not specified. These should be explained in the participant section. It should be stated which statistical reference was used to determine how many people will be included in the study. It should be stated whether a power analysis was performed or whether 5-10 times the number of questions were used as a reference. It should be noted if there were people excluded from the study. Additionally, the flow diagram should be added as a figure.

Response 4:

  • Thank you so much for your valuable suggestions.
  • We agree with your points that further elaborating on this point using new data would be helpful.
  • We added the inclusion and exclusion criteria in subsection 2.2. “Participants and settings” in the revised manuscript. Please see the added text: “Participation in the research presupposed the fulfillment of certain criteria, namely: (a) the participants had to work in the hospital with the professional status of doctor or nurse, (b) the nurses had to be registered nurses (European Qualification Framework - EQF 6), and (c) the participants had to be willing to take part in the research. Exclusion criteria from the study were the following: (a) students in hospitals, (b) practitioner nurses (EQF 5), and (c) health professionals with specialties other than doctors and nurses” in page 3, 1st paragraph, lines: 125-131.
  • We added the statistical reference used to determine how many people will be included in the study, in subsection 2.2. “Participants and settings” in the revised manuscript. Please see the added text: “The minimum sample size for Confirmatory Factor Analysis was calculated equal to 289, according to R package lavaan” in page 3, 1st paragraph, lines: 123-125.
  • In the revised manuscript the above new literature reference is cited as: [23] and is mentioned in the list of references, in page 12, lines: 439-440.
  • In the revised manuscript, in subsection 2.2. “Participants and settings” we added the sentence: “No subjects were excluded from the study as all met the criteria we set”. Please see the added text in page 3, at the end of 1st paragraph, line: 131.

Comments: 5.     Please indicate in the survey description section what the increase or decrease in the score of the survey used means and whether there is a cut-off score.

Response 5:

  • Thank you so much for your valuable suggestions.
  • During the statistical analysis of the research we carried out, there was no cut-off score.

Comments: 6.     Page 5, line 201; “2.6. Ethical Considerations” You do not need to use this heading in the method section. Explain this information in the Study design section. When making an ethical declaration, include in this section the information that the research was conducted in accordance with the Declaration of Helsinki.

Response 6:

  • Thank you so much for your valuable suggestions.
  • We removed subsection 2.6 (original manuscript, page 5, lines: 201-218). In the revised manuscript we moved the corresponding text to section 2.1 “Study Design”, on page 3, 2nd and 3rd paragraphs, lines 101-118.
  • We also included in this section the information that the research was conducted in accordance with the Declaration of Helsinki (revised manuscript, page 3, 3rd paragraph, lines: 115-116).

Comments: 7.     Page 9, line 278-282; “According to the available data we derived from the literature review, the present study is the first in Greece providing a validated digital skills competence instrument for health professionals. Research on the digital skills of employees in various professional fields is constantly gaining ground, but in the hospital environment, it appears to be particularly limited.” There is no need for statements stating the purpose of the study again in the first paragraph of the Discussion section. The beginning of the discussion section should be a paragraph expressing the general result of the study.

Response 7:

  • Thank you so much for your valuable suggestions.
  • We agree with this comment. Therefore, following your recommendations, we removed the 1st paragraph of the discussion section (original manuscript, page 9, lines: 278-282).

Comments: 8.     Page 9, line 277; The discussion section is short and should be improved. Although it has been stated that there are no similar studies, associations can be made with similar studies. Therefore, the discussion should be reviewed from this perspective. Indirect associations should be made.

Response 8:

  • Thank you so much for your valuable suggestions. We agree with your points.
  • We agree with your points that further elaborating on this point using new data would be helpful.
  • We added a new 2nd paragraph with more indirect associations made with similar studies testing measurement tools for digital skills among health professionals.
  • The new paragraph added is in the revised manuscript in section 4. “Discussion”, on page 9, 2nd paragraph, lines: 286-297, with 4 more literature references.
  • In the revised manuscript the above new literature references are cited as: [34],[35], [36] and [37] and are mentioned in the list of references, in page 13, lines: 486-497.

We appreciate the thoughtful and positive comments from the Reviewers. We hope you find the revised manuscript suitable for publication.

We would like to thank you again for your effort, as well as for taking the time to review our manuscript.